

# Efficacy and indications of tonsillectomy in patients with IgA nephropathy: a retrospective study

Yan Li[1], Qi Wan[2], Zhixin Lan[3], Ming Xia[3], Haiyang Liu[3], Guochun Chen[3], Liyu He[3], Chang Wang[3] and Hong Liu[3]

[1] Department of Nephrology, Hunan Provincial People's Hospital, Hunan Normal University, Changsha, Hunan, China

[2] Department of Orthopedic Surgery, The First Affiliated Hospital of Nanchang University, Nanchang, Jiangxi, China

[3] Department of Nephrology, The Second Xiangya Hospital, Central South University, Hunan Key Laboratory of Kidney Disease and Blood Purification, Changsha, Hunan, China

## ABSTRACT

**Background**. The efficacy and indications of tonsillectomy in IgA nephropathy (IgAN) remain uncertain.

**Methods**. We performed a retrospective cohort study of 452 patients with primary IgAN, including 226 patients who received tonsillectomy and 226 controls selected by propensity score matching who had never undergone tonsillectomy. Study outcomes were clinical remission defined as negative hematuria and proteinuria on three consecutive visits over a 6-month period, the endpoint defined as end-stage renal disease or an irreversible 100% increase in serum creatinine from the baseline value. In addition, we further analyzed the critical level of proteinuria in the efficacy of tonsillectomy and the correlation between MEST-C score and tonsillectomy.

**Results**. Up to December 2019, the follow-up period lasted 46 ± 23 months (12–106 months). Kaplan–Meier and multivariate Cox regression analysis revealed that tonsillectomy was beneficial for clinical remission and renal survival. Whether proteinuria was ≤ 1 g/24h or >1 g/24h, the clinical remission and renal survival rates were greater in patients treated with tonsillectomy than without. When the pathological damage was mild or relatively severe, tonsillectomy may be beneficial to clinical remission or renal survival.

**Conclusions**. Tonsillectomy had a favorable effect on clinical remission and delayed renal deterioration in IgAN. In addition to patients with early stage IgAN, it may also be beneficial to IgAN patients with higher levels of proteinuria and relatively severe pathological damage.

## INTRODUCTION

IgA nephropathy (IgAN) is the most widespread type of glomerulonephritis. The incidence in Asian is significantly higher than in European and African (*Li & Yu, 2018*). In China, it accounts for approximately 40%–50% of primary glomerular diseases, and it is the most

Corresponding author
Hong Liu, liuhong618@csu.edu.cn

common cause of uremia (*Li & Liu, 2004*). A cohort of 901 adults from the European Validation Study of the Oxford Classification of IgA Nephropathy (VALIGA) and North American cohorts indicated the 10-year risks of end-stage renal disease (ESRD) or halving of eGFR were 26.8% (*Barbour et al., 2016*). Pacific Asian individuals had a higher risk for ERSD (*Barbour et al., 2013*).

Although the clinical manifestations of IgAN vary greatly, many patients show episodic macroscopic hematuria after upper respiratory tract infections, which suggests a relationship between IgAN and mucosal immunity (*Canetta, Kiryluk & Appel, 2014*; *Rollino, Vischini & Coppo, 2016*). Some studies, almost all retrospective, have examined the therapeutic efficacy of tonsillectomy in IgAN, but the results were inconsistent (*Chen et al., 2007*; *Feehally et al., 2016*; *Hotta et al., 2001*; *Maeda et al., 2012*; *Piccoli et al., 2010*; *Rasche, Schwarz & Keller, 1999*; *Xie et al., 2003*). Cohorts mainly from Japan confirmed the benefits of tonsillectomy on clinical remission or renal survival, mostly in association with steroid pulse therapy. However, several European reports (*Feehally et al., 2016*; *Piccoli et al., 2010*; *Rasche, Schwarz & Keller, 1999*) contradicted these findings. This divergence might be related to the different genetic backgrounds of patients (*Feehally et al., 2016*) and the severity of disease, as well as the small number of cases included in some European researches. Currently, even though the KDIGO clinical practice guideline does not recommend tonsillectomy as part of treatment of IgAN (*Rovin et al., 2021*), tonsillectomy remains a common practice in Japan (*Tomino, 2016*). In China, there are few studies in which a large cohort of IgAN patients with tonsillectomy are followed to assess the effectiveness of tonsillectomy alone.

Although tonsillectomy is a basic operation in otorhinolaryngology, there are some complications associated with it. Post-tonsillectomy hemorrhage occurs in 10–20% of the patients and 0–5% of the patients require hemostasis under general anesthesia (*Kondo et al., 2019*). Therefore, it is beneficial to select patients suitable for tonsillectomy to avoid the risk of poor outcomes. However, indications for tonsillectomy are still unclear. Many factors affect the efficacy of tonsillectomy in IgAN patients, such as urinary findings and grades of renal damage (*Xie et al., 2004*). *Sato et al. (2004)* pointed out that steroid pulse therapy combined with tonsillectomy may be effective in the IgAN patients with a baseline creatinine level of ≤2 mg/dL. However, the critical level of proteinuria at which benefit can be derived from tonsillectomy and the correlation between MEST-C score and tonsillectomy remain uncertain.

In the present research, we conducted a retrospective cohort study of 452 patients with IgAN, comparing the hematuria and proteinuria remission rates and renal survival rates of 226 patients who received tonsillectomy and 226 controls selected by propensity score matching who had never undergone tonsillectomy, paying particular attention to the critical level of proteinuria in the efficacy of tonsillectomy and the correlation between MEST-C score and tonsillectomy.

## MATERIALS & METHODS

### Study design and population

We retrospectively screened the clinical and histologic data and the follow-up data for 2001 patients diagnosed with IgAN by renal biopsy at the Second Xiangya Hospital of Central South University from June 2011 to December 2018. The diagnosis was limited to primary mesangial proliferative glomerulonephritis with deposits of immune complex staining predominantly for IgA in light microscopy and immunofluorescence. Patients combined with systemic diseases such as systemic lupus erythematosus, Henoch-Schönlein purpura, viral hepatitis, diabetes mellitus or malignancy were excluded. Of these 2001 patients, 310 patients underwent tonsillectomy. All of these had a diagnosis of IgAN before tonsillectomy and all cases were done for the purpose of aiding the outcome of IgAN. The exclusion criteria were as follows: (1) age younger than 15 years or older than 60 years (*Hotta et al., 2001*; *Komatsu et al., 2008*), (2) the interval from renal biopsy to tonsillectomy exceeded 6 months, (3) follow-up period less than 12 months. We selected the matched control group from patients with primary IgAN who had never received tonsillectomy by propensity score matching based on age, gender, hematuria score, proteinuria, serum creatinine, serum IgA, mean arterial pressure, MEST-C score and background therapy (steroid/immunosuppressant and RAS inhibitor). Informed consent was waived. This protocol and the application for exemption of informed consent was approved by the Ethics Committee of the Second Xiangya Hospital, Central South University (Approval number: 2019SNK1222000).

### Clinical, laboratory, and pathological data

Baseline clinical and laboratory data at the time of the diagnostic renal biopsy were collected retrospectively. Patient characteristics included age, gender, degree of hematuria and proteinuria, serum creatinine, serum IgA and mean arterial pressure. The results of urinary occult blood (UOB) and urinary protein were converted into scores as follows: (-) to 0, (±) to 0.5, (+) to 1, (2+) to 2 and (3+) to 3.

All pathology slides re-analyzed according to the MEST-C scoring system, proposed by the IgA Nephropathy Classification Working Group (*Trimarchi et al., 2017*). At least two pathologists blinded to patient outcomes confirmed the results. The pathological variables were evaluated as follows: mesangial score ≤0.5 (M0) or >0.5 (M1); endocapillary hypercellularity absent (E0) or present (E1); segmental glomerulosclerosis absent (S0) or present (S1); tubular atrophy/interstitial fibrosis ≤25% (T0), 26%–50% (T1), or >50% (T2); cellular/fibrocellular crescents absent (C0), present in at least 1 glomerulus (C1), in ≥25% of glomeruli (C2).

### Treatment

The standard indications for tonsillectomy were repeated episodes of tonsillitis three or more times a year, recurring gross hematuria during tonsillitis or chronic tonsillitis with pus in tonsillar crypt (*Maeda et al., 2012*). The diagnosis of tonsillitis or chronic tonsillitis were made by professional otolaryngologists. In addition, for some patients who wanted

to obtain clinical remission, tonsillectomy could be performed after signing the informed consent, regardless of whether the patient met the above criteria.

Drug therapies included steroid (oral or pulse therapy), immunosuppressant, and RAS inhibitor (angiotensin-converting enzyme inhibitors and angiotensin receptor blockers) during follow-up. Main indications for steroid and/or immunosuppressant therapy were if patients had symptoms of progressive diseases such as erythrocyte cast, high-grade proteinuria, acute renal failure, or advanced histological findings including endocapillary proliferation, glomerular sclerosis, interstitial infiltration, or crescent formation. In oral steroid therapy, 6-month regime of oral prednisone starting with 0.8–1 mg/kg/d for 2 months and then reduced by 0.2 mg/kg/d per month for the next 4 months. In pulse steroid therapy, 500 mg/d of methylprednisolone intravenously for 3 days each at months 1, 3, and 5, followed by oral steroid 0.5 mg/kg prednisone on alternate days for 6 months. All drugs were used according to the KDIGO clinical practice guideline (*Rovin et al., 2021*).

## Outcomes

The definition of negative hematuria was urinary occult blood ranging from (-) to ($\pm$). The criterion for negative proteinuria was urinary protein ranging from (-) to ($\pm$) or less than 150 mg/24h. Hematuria or proteinuria remission was defined as negative hematuria or proteinuria on three consecutive visits over a 6-month period, and clinical remission referred to remission of both hematuria and proteinuria (*Suzuki et al., 2014*). The endpoint of this study was a composite of ESRD or an irreversible 100% increase in serum creatinine from the baseline value.

## Statistical analysis

Normally distributed data were expressed as mean $\pm$ SD, and non-normally distributed data were expressed as medians (Q25, Q75). For categorical variables, the data were presented as counts and percentages. Differences between the groups were compared using unpaired $t$-test, Mann–Whitney $U$ test or chi-square test as appropriate. Kaplan–Meier analysis and log-rank test were used to assess the efficacy and indications of tonsillectomy on clinical remission and cumulative renal survival rates. In addition, we used the Cox proportional hazards model to assess the impact of multiple covariates for clinical remission or renal survival. All variables used in univariate and multivariate analyses were expressed in categorical or quantitative forms. Gender (male/female), hematuria (yes/no), M (M0/M1), E (E0/E1), S (S0/S1), T (T0/T1+T2), C (C0/C1+C2), tonsillectomy (yes/no), steroid/immunosuppressant (yes/no), RAS inhibitor (yes/no) were regarded as categorical variables and were expressed using a binary scale, coded as 0/1. Age, proteinuria, serum creatinine and mean arterial pressure were regarded as quantitative variables. We calculated coefficient and 95% confidence intervals for each hazard ratio. In all calculations, $P<0.05$ was considered statistically significant. Propensity score matching and statistical analyses were performed by SPSS 24.0 (SPSS Inc., Chicago, IL, USA).

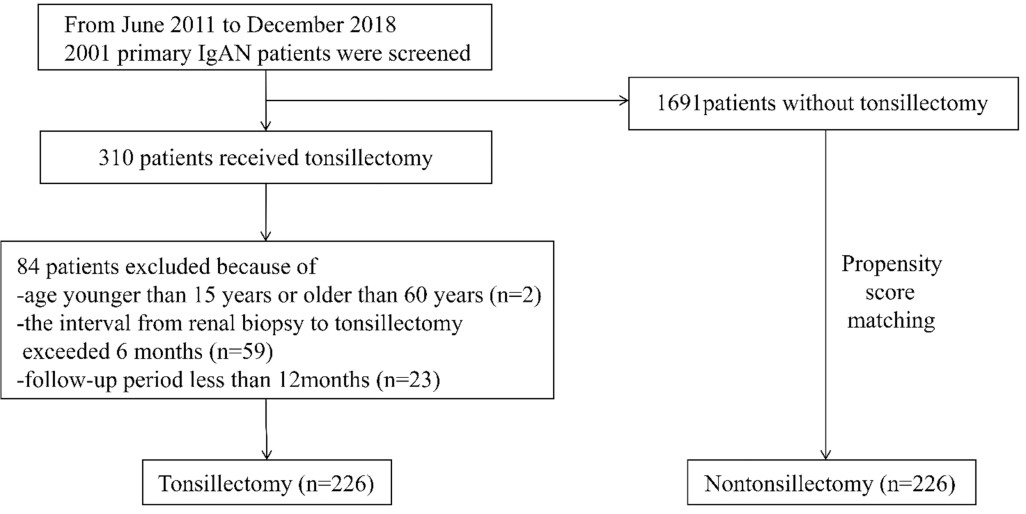

**Figure 1  Study recruitment/inclusion flow-chart.**

## RESULTS

### Baseline characteristics

From June 2011 to December 2018, a total of 310 primary IgAN patients received tonsillectomy as an additional treatment. Of these 310 patients, 226, who met the inclusion criteria were enrolled. 226 controls matched for age, gender, hematuria score, proteinuria, serum creatinine, serum IgA, mean arterial pressure, MEST-C score and background therapy (steroid/immunosuppressant and RAS inhibitor) were selected by propensity score matching (Fig. 1). The mean duration of follow-up for all participants was 46 ± 23 months (range, 12 to 106 months). The mean follow-up periods in the tonsillectomy and nontonsillectomy groups were 45 ± 23 months and 47 ± 22 months, respectively. Baseline clinical, laboratory, pathological data and background therapy were not statistically different between the tonsillectomy and nontonsillectomy groups (Table 1). Among the patients who underwent tonsillectomy, 154 had chronic tonsillitis, 162 had a history of recurrent tonsillitis, 85 had recurring gross hematuria during tonsillitis, and six had no documented tonsil anomaly. In the nontonsillectomy group, 142 had chronic tonsillitis, 151 had a history of recurrent tonsillitis, and 79 had recurring gross hematuria during tonsillitis, which showed no statistical difference from the tonsillectomy group (Table 1).

### Hematuria and proteinuria remission

After treatment, hematuria and proteinuria were significantly relieved, and the remission effects of hematuria and proteinuria in tonsillectomy group were higher than those in nontonsillectomy group (Table S1). Of the 226 patients with tonsillectomy, 125 achieved hematuria remission, 138 got proteinuria remission, and 98 had clinical remission, whereas of the 226 patients without tonsillectomy, 97 achieved hematuria remission, 116 obtained proteinuria remission, and 67 had clinical remission. The probabilities of hematuria remission, proteinuria remission and clinical remission among all patients according to

**Table 1  Baseline characteristics of the patients with or without tonsillectomy.**

| Characteristic | Tonsillectomy N = 226 | Nontonsillectomy N = 226 | P-value |
|---|---|---|---|
| Age, years | 30 ± 8 | 29 ± 9 | 0.726 |
| Gender, male/female | 91/135 | 83/143 | 0.439 |
| Chronic tonsillitis | 154 (68%) | 142 (63%) | 0.235 |
| History of recurrent tonsillitis | 162 (72%) | 151 (67%) | 0.262 |
| Recurring gross hematuria during tonsillitis | 85 (38%) | 79 (35%) | 0.557 |
| Hematuria score, 0/0.5/1/2/3 | 5/17/35/100/69 | 4/25/39/97/61 | 0.665 |
| Proteinuria, g/24h | 0.52 (0.32, 1.17) | 0.50 (0.24, 1.05) | 0.098 |
| Serum creatinine, $\mu$moI/L | 70 (57, 89) | 71 (58, 89) | 0.909 |
| Serum IgA, mg/dl | 2.84 ± 1.02 | 2.78 ± 1.03 | 0.621 |
| Systolic blood pressure, mmHg | 125 ± 15 | 124 ± 15 | 0.625 |
| Diastolic blood pressure, mmHg | 80 ± 11 | 81 ± 12 | 0.528 |
| Mean arterial pressure, mmHg | 95 ± 11 | 95 ± 12 | 0.828 |
| Oxford Classification | | | |
|     M 0/1 | 95/131 | 101/125 | 0.569 |
|     E 0/1 | 223/3 | 223/3 | 1.000 |
|     S 0/1 | 48/178 | 58/168 | 0.267 |
|     T 0/1/2 | 175/45/6 | 177/46/3 | 0.676 |
|     C 0/1/2 | 163/61/2 | 174/50/2 | 0.449 |
| Background therapy | | | |
|     Steroid/Immunosuppressant | 129 (57%) | 123 (54%) | 0.570 |
|     RAS inhibitor | 162 (72%) | 154 (68%) | 0.412 |

Notes.
Values are expressed as mean ± SD, medians (Q25, Q75) or % and compared using unpaired $t$-test, Mann–Whitney $U$ test or chi-square test, respectively.

tonsillectomy were evaluated by Kaplan–Meier analysis. There were significant positive correlations between tonsillectomy and the remission of urinary abnormalities, and the rates of hematuria remission, proteinuria remission and clinical remission in tonsillectomy group were much higher than those in nontonsillectomy group (Figs. 2A–2C).

## Renal survival

Among 226 patients with tonsillectomy, 4 (1.8%) reached the endpoint and required dialysis therapy, while among 226 patients without tonsillectomy, 16 (7.1%) reached the endpoint and seven needed dialysis therapy. We performed a Kaplan–Meier analysis comparing the renal survival for all patients according to tonsillectomy. The mean renal survival time for patients with tonsillectomy and nontonsillectomy were 104 months (standard error 1.0) and 92 months (standard error 2.0), respectively. The estimated renal survival rate at 8 years in tonsillectomy patients was 97%, but was 79% in nontonsillectomy patients. In the log-rank test, there was a significant difference in renal survival rates between the two groups ($P = 0.012$, Fig. 2D).

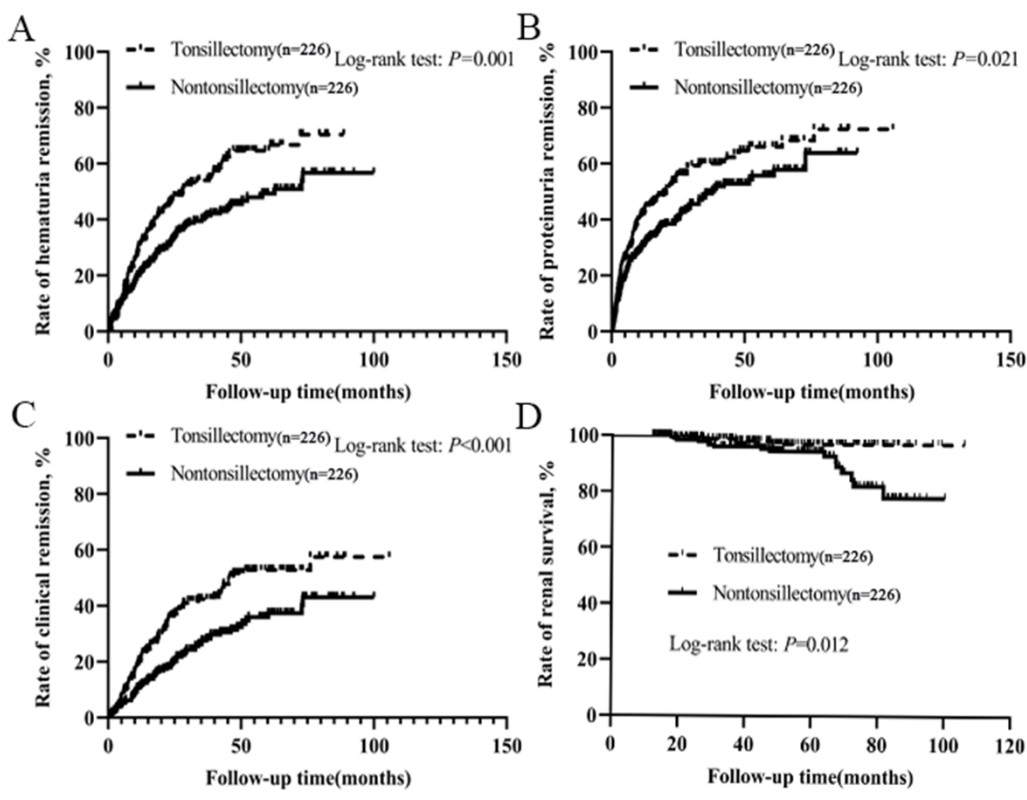

**Figure 2** (A–C) Kaplan–Meier analysis of hematuria remission, proteinuria remission and clinical remission among IgAN patients according to tonsillectomy. (D) Kaplan–Meier analysis of renal survival for IgAN patients according to tonsillectomy.

## Cox regression analysis

The known risk factors for clinical remission or renal survival were selected as imperative independent variables in the model. As shown in Table 2, hematuria ($P = 0.015$), proteinuria ($P = 0.017$), T score ($P = 0.005$) and tonsillectomy ($P < 0.001$) were independently associated with clinical remission in multivariate Cox regression analysis. The HR of hematuria, proteinuria and T score were all <1, suggesting that a high amount of proteinuria and hematuria, tubular atrophy/interstitial fibrosis at biopsy were related to decreased clinical remission. The HR of tonsillectomy was >1, indicating that tonsillectomy was linked to increased clinical remission. Similarly, a high amount of proteinuria, elevated serum creatinine and tubular atrophy/interstitial fibrosis at biopsy were independent risk factors for renal survival, while tonsillectomy was independently associated with a greater renal survival (Table 3).

## Patients who did not receive steroid and immunosuppressant

To eliminate the therapeutic effect of steroid and immunosuppressant, we conducted a separate analysis for patients who did not receive steroid and immunosuppressant. Characteristics of the patients without steroid and immunosuppressive therapy were presented in Table 4. Baseline clinical, laboratory, pathological data and RAS inhibitor

**Table 2** Univariate and multivariate analysis of factors that contribute to clinical remission in IgA nephropathy.

| Variable | Univariate Analysis | | | Multivariate Analysis | | |
|---|---|---|---|---|---|---|
| | HR | 95% CI | *P*-value | HR | 95% CI | *P*-value |
| Age (per decade) | 1.02 | (0.84–1.22) | 0.872 | 1.11 | (0.90–1.37) | 0.337 |
| Gender (male *vs* female) | 0.80 | (0.59–1.09) | 0.158 | 0.71 | (0.48–1.05) | 0.088 |
| Hematuria (*vs* no hematuria) | 0.53 | (0.36–0.80) | 0.003[*] | 0.59 | (0.38–0.90) | 0.015[*] |
| Proteinuria (g/24h) | 0.71 | (0.57–0.89) | 0.003[*] | 0.74 | (0.57–0.95) | 0.017[*] |
| Serum creatinine (per 10 $\mu$moI/L) | 0.96 | (0.91–1.02) | 0.145 | 0.98 | (0.89–1.07) | 0.585 |
| Mean arterial pressure (per 10 mmHg) | 0.98 | (0.86–1.11) | 0.743 | 1.06 | (0.92–1.23) | 0.426 |
| M (M0 *vs* M1) | 0.78 | (0.58–1.06) | 0.112 | 0.74 | (0.54–1.03) | 0.073 |
| E (E0 *vs* E1) | 0.47 | (0.07–3.33) | 0.446 | 0.60 | (0.08–4.73) | 0.630 |
| S (S0 *vs* S1) | 1.32 | (0.90–1.94) | 0.152 | 1.51 | (0.99–2.28) | 0.053 |
| T (T0 *vs* T1+T2) | 0.49 | (0.32–0.75) | 0.001[*] | 0.50 | (0.31–0.81) | 0.005[*] |
| C (C0 *vs* C1+C2) | 1.04 | (0.75–1.45) | 0.797 | 1.27 | (0.89–1.83) | 0.188 |
| Tonsillectomy | 1.75 | (1.28–2.39) | <0.001[*] | 1.77 | (1.29–2.42) | <0.001[*] |
| Steroid/Immunosuppressant | 0.91 | (0.67–1.24) | 0.550 | 1.00 | (0.72–1.39) | 0.990 |
| RAS inhibitor | 0.99 | (0.71–1.40) | 0.965 | 0.99 | (0.68–1.44) | 0.974 |

**Notes.**

*HR*, hazard ratio; *CI*, confidence interval.
*Statistically significant.

therapy were not statistically different between the two groups. Kaplan–Meier analysis demonstrated that the rates of clinical remission and renal survival in tonsillectomy group were much higher than those in nontonsillectomy group (Fig. 3).

## Patients who had received steroid/immunosuppressant

We also conducted a separate analysis for patients who had received steroid/immunosuppressant. As shown in Table 5, baseline clinical, laboratory, pathological data and RAS inhibitor therapy were not statistically different between the two groups. Kaplan–Meier analysis indicated that the clinical remission rate of tonsillectomy group was much higher than that of nontonsillectomy group (Fig. 4). Among 129 patients with tonsillectomy, three (2.3%) reached the endpoint, while among 123 patients without tonsillectomy, nine (7.3%) reached the endpoint. Kaplan–Meier analysis suggested a trend towards improved renal survival with tonsillectomy, although the log-rank test showed no significant difference between the two groups ($P = 0.079$).

## Critical level of proteinuria in the efficacy of tonsillectomy

In order to further understand the effect of proteinuria on the efficacy of tonsillectomy, we analyzed the graded effect of proteinuria on clinical remission and renal survival. Patients were divided into two groups based on their proteinuria: $\leq 1$ and $>1$ g/24h. In different proteinuria grades, the background therapy (steroid/immunosuppressant and RAS inhibitor) was not different for patients with and without tonsillectomy (Table S2). As illustrated in Fig. 5, the clinical remission rates of tonsillectomy group and nontonsillectomy

**Table 3** Univariate and multivariate analysis of factors that contribute to renal survival in IgA nephropathy.

| Variable | Univariate analysis | | | Multivariate analysis | | |
|---|---|---|---|---|---|---|
| | HR | 95% CI | *P*-value | HR | 95% CI | *P*-value |
| Age (per decade) | 1.12 | (0.68–1.84) | 0.654 | 1.18 | (0.67–2.10) | 0.567 |
| Gender (male *vs* female) | 0.84 | (0.35–2.02) | 0.688 | 1.51 | (0.54–4.17) | 0.430 |
| Hematuria (*vs* no hematuria) | 2.94 | (0.39–22.01) | 0.293 | 1.25 | (0.15–10.32) | 0.833 |
| Proteinuria (g/24h) | 1.88 | (1.50–2.36) | <0.001[*] | 1.84 | (1.21–2.80) | 0.004[*] |
| Serum creatinine (per 10 $\mu$mol/L) | 1.31 | (1.22–1.40) | <0.001[*] | 1.23 | (1.07–1.40) | 0.003[*] |
| Mean arterial pressure (per 10 mmHg) | 1.37 | (1.02–1.84) | 0.036[*] | 1.01 | (0.66–1.53) | 0.979 |
| M (M0 *vs* M1) | 0.79 | (0.32–1.90) | 0.591 | 0.75 | (0.26–2.13) | 0.585 |
| E (E0 *vs* E1) | 4.89 | (0.65–36.64) | 0.123 | 1.16 | (0.06–20.98) | 0.922 |
| S (S0 *vs* S1) | 1.92 | (0.56–6.57) | 0.298 | 1.26 | (0.26–6.22) | 0.775 |
| T (T0 *vs* T1+T2) | 6.84 | (3.46–13.53) | <0.001[*] | 4.00 | (1.81–8.86) | 0.001[*] |
| C (C0 *vs* C1+C2) | 1.74 | (0.75–4.06) | 0.198 | 0.94 | (0.32–2.77) | 0.904 |
| Tonsillectomy | 0.27 | (0.09–0.81) | 0.019[*] | 0.28 | (0.08–0.96) | 0.042[*] |
| Steroid/Immunosuppressant | 0.99 | (0.40–2.43) | 0.981 | 0.60 | (0.20–1.77) | 0.355 |
| RAS inhibitor | 0.80 | (0.31–2.10) | 0.655 | 0.87 | (0.24–3.09) | 0.825 |

**Notes.**

*HR*, hazard ratio; *CI*, confidence interval.

[*]Statistically significant.

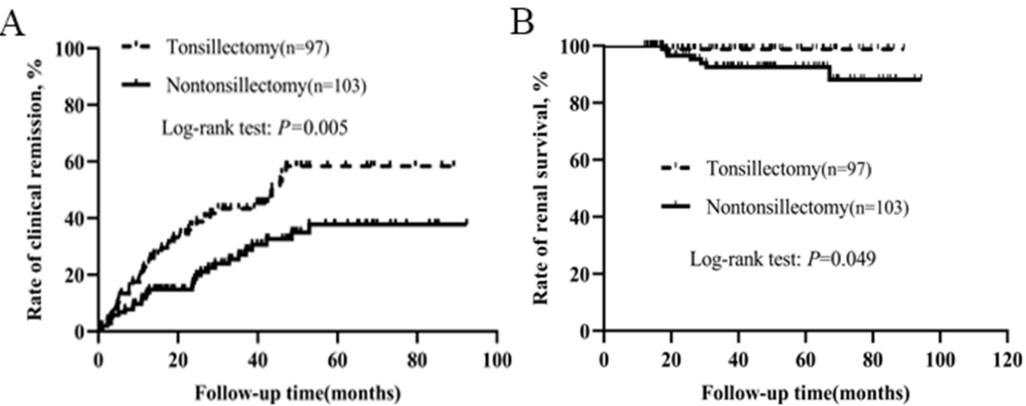

**Figure 3** Kaplan–Meier analysis of clinical remission and renal survival in IgAN patients who did not receive steroid and immunosuppressant according to tonsillectomy.

group were significantly different not only when proteinuria was ≤1 g/24h, but also when proteinuria was >1 g/24h.

The graded effects of proteinuria on renal survival in tonsillectomy and nontonsillectomy patients were illustrated in Fig. 6. Whether proteinuria was ≤1 g/24h or >1 g/24h, the renal survival rates were greater in patients treated with tonsillectomy than without.

**Table 4  Baseline characteristics of the patients with or without tonsillectomy who did not receive steroid and immunosuppressant.**

| Characteristic | Tonsillectomy N = 97 | Nontonsillectomy N = 103 | P-value |
|---|---|---|---|
| Age, years | 29 ± 8 | 30 ± 8 | 0.787 |
| Gender, male/female | 35/62 | 39/64 | 0.794 |
| Chronic tonsillitis | 73 (75%) | 66(64%) | 0.086 |
| History of recurrent tonsillitis | 76 (78%) | 71 (69%) | 0.131 |
| Recurring gross hematuria during tonsillitis | 39 (40%) | 36 (35%) | 0.443 |
| Hematuria score, 0/0.5/1/2/3 | 3/6/15/49/24 | 1/13/22/46/21 | 0.285 |
| Proteinuria, g/24h | 0.50 (0.22, 1.13) | 0.48 (0.24, 0.98) | 0.599 |
| Serum creatinine, $\mu$moI/L | 66 (54, 86) | 71 (55, 84) | 0.753 |
| Serum IgA, mg/dl | 2.90 ± 1.03 | 2.81 ± 1.00 | 0.584 |
| Systolic blood pressure, mmHg | 125 ± 16 | 123 ± 15 | 0.528 |
| Diastolic blood pressure, mmHg | 79 ± 12 | 80 ± 12 | 0.697 |
| Mean arterial pressure, mmHg | 94 ± 12 | 94 ± 12 | 0.984 |
| Oxford Classification | | | |
|    M 0/1 | 49/48 | 48/55 | 0.580 |
|    E 0/1 | 96/1 | 103/0 | 0.302 |
|    S 0/1 | 26/71 | 29/74 | 0.831 |
|    T 0/1/2 | 80/15/2 | 85/17/1 | 0.884 |
|    C 0/1/2 | 70/26/1 | 85/16/2 | 0.098 |
| Background therapy | | | |
|    RAS inhibitor | 57(59%) | 55(53%) | 0.445 |

**Notes.**

Values are expressed as mean ± SD, medians (Q25, Q75) or % and compared using unpaired $t$-test, Mann–Whitney $U$ test or chi-square test, respectively.

## Correlation between MEST-C score and tonsillectomy

To understand the correlation between histopathologic findings and tonsillectomy, Kaplan–Meier analysis was used to compare the clinical remission rates and renal survival rates of 226 patients who received tonsillectomy and 226 controls who had never undergone tonsillectomy under different MEST-C scores. In different MEST-C scores, the background therapy (steroid/immunosuppressant and RAS inhibitor) was not different for patients with and without tonsillectomy (Table S3). We did not perform Kaplan–Meyer analysis on histological lesions with low frequency (E1, T2, C2). As illustrated in Figs. S1 and S2, when the pathological damage was mild or relatively severe, tonsillectomy may be beneficial to clinical remission or renal survival.

## DISCUSSION

Since the mechanisms of onset and progression of IgAN remain obscure, specific treatment has not been established (*Floege, 2011*). Although tonsillectomy has been suggested as a possible treatment modality in Japan, up to now there has been little information in China about the efficacy and indications of tonsillectomy in IgAN patients. In this historical cohort study among 452 patients with primary IgAN confirmed by biopsy, we

**Table 5 Baseline characteristics of the patients with or without tonsillectomy who had received steroid/immunosuppressant.**

| Characteristic | Tonsillectomy N = 129 | Nontonsillectomy N = 123 | P-value |
|---|---|---|---|
| Age, years | 30 ± 7 | 29 ± 9 | 0.481 |
| Gender, male/female | 56/73 | 44/79 | 0.215 |
| Chronic tonsillitis | 81 (63%) | 76 (62%) | 0.870 |
| History of recurrent tonsillitis | 86 (67%) | 80 (65%) | 0.786 |
| Recurring gross hematuria during tonsillitis | 46 (36%) | 43 (35%) | 0.908 |
| Hematuria score, 0/0.5/1/2/3 | 2/11/20/51/45 | 3/12/17/51/40 | 0.959 |
| Proteinuria, g/24h | 0.60 (0.36, 1.23) | 0.50 (0.24, 1.10) | 0.099 |
| Serum creatinine, $\mu$moI/L | 75 (59, 98) | 71 (60, 93) | 0.729 |
| Serum IgA, mg/dl | 2.80 ± 1.01 | 2.76 ± 1.05 | 0.780 |
| Systolic blood pressure, mmHg | 125 ± 14 | 125 ± 14 | 0.963 |
| Diastolic blood pressure, mmHg | 81 ± 11 | 82 ± 13 | 0.586 |
| Mean arterial pressure, mmHg | 95 ± 11 | 96 ± 12 | 0.724 |
| Oxford Classification | | | |
| M 0/1 | 46/83 | 53/70 | 0.227 |
| E 0/1 | 127/2 | 120/3 | 0.613 |
| S 0/1 | 22/107 | 29/94 | 0.198 |
| T 0/1/2 | 95/30/4 | 92/29/2 | 0.828 |
| C 0/1/2 | 93/35/1 | 89/34/0 | 0.963 |
| Background therapy | | | |
| RAS inhibitor | 105(81%) | 99(80%) | 0.854 |

**Notes.**
Values are expressed as mean ± SD, medians (Q25, Q75) or % and compared using unpaired *t*-test, Mann–Whitney *U* test or chi-square test, respectively.

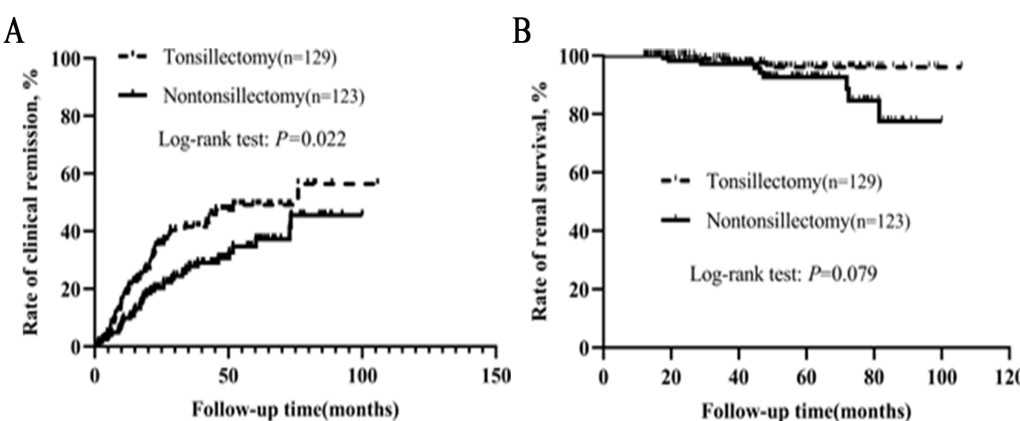

**Figure 4** (A–B) Kaplan–Meier analysis of clinical remission and renal survival in IgAN patients who had received steroid/immunosuppressant according to tonsillectomy.

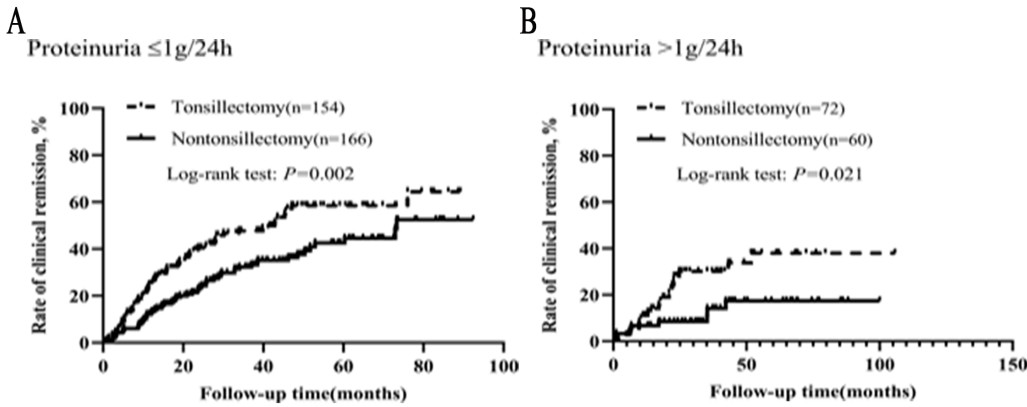

**Figure 5** (A–B) Kaplan–Meier analysis of clinical remission between the tonsillectomy and nontonsillectomy groups under different proteinuria levels at onset.

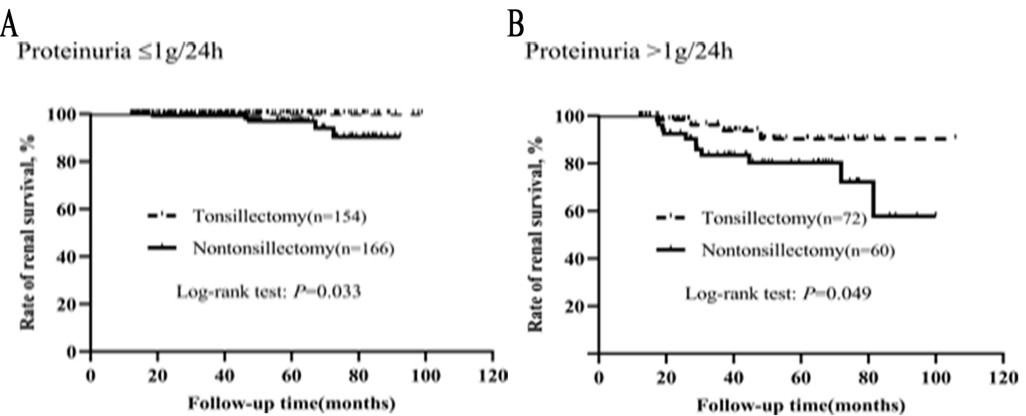

**Figure 6** (A–B) Kaplan–Meier analysis of renal survival between the tonsillectomy and nontonsillectomy groups under different proteinuria levels at onset.

demonstrated that tonsillectomy was significantly associated with clinical remission and delayed renal deterioration during the follow-up. Given that initial proteinuria and MEST-C score were closely related to the prognosis of IgAN and the efficacy of tonsillectomy, we further analyzed the critical level of proteinuria which was effective in tonsillectomy and the correlation between MEST-C score and tonsillectomy. The results showed that whether proteinuria was ≤1 g/24h or >1 g/24h, the clinical remission and renal survival rates were greater in patients treated with tonsillectomy than without. When the pathological damage was mild or relatively severe, tonsillectomy may be beneficial to clinical remission or renal survival.

Several studies have examined the therapeutic efficacy of tonsillectomy in IgAN, but the results were inconsistent (*Chen et al., 2007*; *Feehally et al., 2016*; *Hotta et al., 2001*; *Maeda et al., 2012*; *Piccoli et al., 2010*; *Rasche, Schwarz & Keller, 1999*; *Xie et al., 2003*). *Hotta et al. (2001)* conducted a retrospective investigation of IgAN patients from Japan, with a median

observation period of 75 months. They found that combination therapy of steroids and tonsillectomy was superior to steroids alone in inducing clinical remission. By multivariate Cox regression analysis, tonsillectomy and steroid pulse therapy had significant impacts on clinical remission. Similar results were described in a report from China that the clinical remission rate in IgAN patients with tonsillectomy was higher than that in patients without tonsillectomy (*Chen et al., 2007*). However, they failed to demonstrate that tonsillectomy can prevent renal deterioration. Two retrospective observations from Japan indicated that tonsillectomy had a favorable effect on renal survival in an early and mild stage of IgAN (*Maeda et al., 2012*; *Xie et al., 2003*). Nevertheless, several European reports (*Feehally et al., 2016*; *Piccoli et al., 2010*; *Rasche, Schwarz & Keller, 1999*) contradicted these findings. This may be because in these European studies, two studies enrolled a small number of cases (*Feehally et al., 2016*; *Rasche, Schwarz & Keller, 1999*), one study patients had a relatively advanced IgAN (*Rasche, Schwarz & Keller, 1999*), and one study outcome was defined as the progression from CKD stage 1–2 to stage 3 or higher (*Piccoli et al., 2010*). This divergence may also be related to the different genetic backgrounds of patients (*Feehally et al., 2016*). Our research demonstrated that tonsillectomy could not only ameliorate urinary findings but also postpone renal deterioration in a large number of IgAN patients from China.

Otolaryngologically, the standard indications for tonsillectomy were repeated episodes of tonsillitis three or more times a year, recurring gross hematuria during tonsillitis or chronic tonsillitis with pus in tonsillar crypt (*Maeda et al., 2012*).For patients with IgAN, the indications for tonsillectomy are to date still unclear. Urinary findings and grades of renal damage have effect on the efficacy of tonsillectomy in IgAN patients (*Xie et al., 2004*). In addition, proteinuria has been proved to be the most important clinical predictors of progression for IgAN (*Berthoux et al., 2011*; *Chen et al., 2018*) and a marked reduction of proteinuria improved the long-term renal survival (*Le et al., 2012*; *Reich et al., 2007*). The value of the Oxford Classification in predicting long-term outcomes of IgAN has also been validated many times since its first publication in 2009 and was updated in 2017 (*Markowitz, 2017*; *Trimarchi et al., 2019*). In this research, whether proteinuria was ≤1 g/24h or >1 g/24h, the clinical remission rates and renal survival rates of tonsillectomy group were significantly higher than those of nontonsillectomy group. When the pathological damage was mild or relatively severe, tonsillectomy may be beneficial to clinical remission or renal survival. Besides, *Sato et al. (2004)* pointed out that tonsillectomy may not change the renal outcome even if combined with steroid therapy when the baseline serum creatinine was >2 mg/dL. These results indicated that in addition to early stage IgAN patients, tonsillectomy may also be beneficial to IgAN patients with higher levels of proteinuria and relatively severe pathological damage. Some reports have been made in Japan, where tonsillectomy was mainly indicated for mild IgAN (*Maeda et al., 2012*; *Xie et al., 2003*), but they did not give the clear definition of mild IgAN. For patients with relatively advanced IgAN, *Rasche, Schwarz & Keller (1999)* conducted a study in which 55% of patients had hypertension, 35% had serum creatinine >150 mmol/L, and 62% had proteinuria >1.5 g/24h, they found that tonsillectomy did not delay renal deterioration in these patients within a median follow-up period of 2.3 years. This diversity may be related to the different genetic backgrounds

of patients, the short follow-up period and the small number of cases included in this European research.

The mechanism regarding how tonsillectomy improves renal survival in IgAN patients has not been fully elucidated. The association of episodic macroscopic haematuria with upper respiratory or gastrointestinal tract infections leads to the hypothesis that IgAN is linked to abnormal mucosal immunity (*Canetta, Kiryluk & Appel, 2014*; *Rollino, Vischini & Coppo, 2016*). It has been reinforced by genome-wide association studies in IgAN, which have identified susceptibility loci in genes that are directly related to mucosal immunity (*Gharavi et al., 2011*). It is suggested that the glomerular mesangial deposits of polymeric IgA are due to the response of a mucosal immune system to environmental pathogens (*Coppo, 2010*). Mucosal immunity works through recognizing pathogen-associated molecular patterns by Toll-like receptors (TLRs). Stimulation of B cells in mucosal lamina propria by TLR-9 ligands containing the CpG-oligodeoxynucleotide induces polyclonal activation of B cells, class switching, and IgA production (*Blaas et al., 2009*). Ligation of B-cell TLR-4 by bacterial lipopolysaccharide leads to methylation of the COSMC gene, resulting in decreased activity of C1GalT1 and undergalactosylation of IgA1 (*Qin et al., 2008*). Polymeric IgA molecules deposited in the mesangium in IgAN are mostly of galactose-deficient IgA1 (*Suzuki et al., 2011*), and tonsillar IgA closely resembles mesangial IgA (*Barratt & Tang, 2018*; *Muto et al., 2017*). Studies show that both the number and relative percentage of IgA-bearing cells are significantly increased in the tonsils of IgAN patients than non-IgAN patients (*Meng et al., 2015*; *Meng et al., 2016*). Although tonsillar IgA-bearing cells typically produce IgA1 (*Brandtzaeg, 2010*), this subclass is even more predominant in the tonsils extracted from IgAN patients compared with from non-IgAN patients (*Meng et al., 2016*). Galactose-deficient IgA1 is found in the tonsil of patients with IgAN (*Horie et al., 2003*) and the pathogenic IgA1 in IgAN is partially derived from tonsil (*Huang et al., 2010*). Moreover, altered glycosylation can reduce the clearance of IgA1 molecules by the liver and increase the binding of IgA1 to the glomerular mesangium (*Bournia & Tektonidou, 2015*; *Novak et al., 2002*). These two kinds of mechanisms interact and form a vicious circle, leading to the occurrence of IgAN. Tonsillectomy can not only remove the infectious pathogens in the pharynx, but also be considered as an easy mean to reduce lymphoid tissue actively recruiting activated IgA-producing cells. This explains partly why tonsillectomy is beneficial for reducing serum IgA level and mesangial IgA deposits in IgAN (*Komatsu et al., 2008*).

Emerging evidence supports the role of mucosal immune system in IgAN pathogenesis. Mucosa-associated lymphoid tissue (MALT) is situated along the surfaces of all mucosal tissues. Its most well-known representatives are gut-associated lymphoid tissue (GALT), nasopharynx-associated lymphoid tissue (NALT) and bronchus-associated lymphoid tissue (BALT). Tonsillectomy can remove pharynx-associated lymphoid tissue, thereby ameliorating renal lesions in IgAN. Recently, a series of studies have focused on the modulation of GALT in IgAN. Dysregulation of the interplay between intestinal immunity, microbiota and diet can lead to the production of mis-galactosylated IgA (*Monteiro, 2018*). Patients with IgAN have an increased risk of celiac disease (4% *vs* 0.5–1%) and in celiac patients, there is an increased risk of IgAN (0.026% *vs* 0.008%) (*Coppo, 2018*). A

genome-wide association study reporting that most loci found at risk for developing IgAN were also related to inflammatory bowel diseases, the maintenance of the intestinal barrier and the regulation of GALT's response to intestinal pathogens (*Kiryluk et al., 2014*). Among patients with IgAN, comorbid inflammatory bowel disease elevates the risk of progression to ESRD (*Rehnberg et al., 2021*). In IgAN patients, enteric budesonide targeting the ileocecal region can ameliorate proteinuria (*Fellström et al., 2017*). Early treatment of humanized mice with a gluten-free diet can prevent mesangial IgA1 deposits and hematuria (*Papista et al., 2015*).

There are several limitations in our study. First, it was a retrospective cohort study despite the baseline data and endpoints were precise because of regular follow-up after renal biopsy and accurate records provided by the same team. Although randomized prospective controlled trials are very important, it is hard to randomize patients to tonsillectomy because tonsillectomy is a surgical operation and its arbitrary application is accompanied by ethical issues. Second, the follow-up period was relatively short so that the number of cases reaching the endpoint was small. To elucidate whether tonsillectomy can protect IgAN patients from progressive renal deterioration under different conditions, further large-scale clinical study with a long follow-up period is needed. Third, all participants in this study were Chinese and the favorable results of tonsillectomy in IgAN may have a relation with ethnic factors. Lastly, six patients in the tonsillectomy group had no indications for tonsillectomy, and they just wanted it to obtain clinical remission. Although the baseline clinical, laboratory, pathological data and background therapy were not statistically different between the tonsillectomy and nontonsillectomy groups, there was a potential bias in the results because patients seeking more treatments were more likely to have more favorable outcomes.

## CONCLUSIONS

In conclusion, tonsillectomy had a favorable effect on clinical remission and delayed renal deterioration in IgAN. In addition to patients with early stage IgAN, it may also be beneficial to IgAN patients with higher levels of proteinuria and relatively severe pathological damage.

### Funding
This work was supported by the National Natural Science Foundation of China (No. 82070737) and the Research Foundation of Jiangxi Provincial Education Department (No. GJJ200224). The funders had no role in study design, data collection and analysis, decision to publish, or preparation of the manuscript.

### Grant Disclosures
The following grant information was disclosed by the authors:
National Natural Science Foundation of China: 82070737.
Research Foundation of Jiangxi Provincial Education Department:  GJJ200224.

## Competing Interests

The authors declare there are no competing interests.

## Author Contributions

- Yan Li conceived and designed the experiments, performed the experiments, analyzed the data, prepared figures and/or tables, authored or reviewed drafts of the article, and approved the final draft.
- Qi Wan performed the experiments, prepared figures and/or tables, authored or reviewed drafts of the article, and approved the final draft.
- Zhixin Lan performed the experiments, prepared figures and/or tables, authored or reviewed drafts of the article, and approved the final draft.
- Ming Xia analyzed the data, prepared figures and/or tables, and approved the final draft.
- Haiyang Liu analyzed the data, prepared figures and/or tables, and approved the final draft.
- Guochun Chen conceived and designed the experiments, authored or reviewed drafts of the article, and approved the final draft.
- Liyu He conceived and designed the experiments, authored or reviewed drafts of the article, and approved the final draft.
- Chang Wang conceived and designed the experiments, authored or reviewed drafts of the article, and approved the final draft.
- Hong Liu conceived and designed the experiments, performed the experiments, analyzed the data, prepared figures and/or tables, authored or reviewed drafts of the article, and approved the final draft.

## Human Ethics

The following information was supplied relating to ethical approvals (i.e., approving body and any reference numbers):

This protocol was approved by the Ethics Committee of the Second Xiangya Hospital, Central South University (Approval number: 2019SNK1222000).

## Data Availability

The raw measurements are available as a Supplementary File.

## Supplemental Information

Supplemental information for this article can be found online at http://dx.doi.org/10.7717/peerj.14481#supplemental-information.

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
