# Peer review of "Efficacy and indications of tonsillectomy in patients with IgA nephropathy: a retrospective study"

_PeerJ, doi:10.7717/peerj.14481_

## Round 0.1 · original submission · Major Revisions

The described study needs some major revisions in regard to statistical analysis in order to assure enough statistical power to the obtained results. Please carefully refer to reviewers' comments for this. I would especially underline the need to: clearly explain how the propensity-matched analysis was performed, to re-organize subgroups of subjects to have comparable numbers (suggestions were offered by reviewer 2 and reviewer 3), to consider immunosuppression therapy (see reviewer 1 comments), to avoid references to specific pathological score. Depending on the results from the new analysis, the Results and Discussion sections may be modified.

In addition, the Discussion section should be enriched with the points to address suggested by reviewer 1.

Reviewer 1 ·

Basic reporting

I have reviewed the manuscript "Efficacy and indications of tonsillectomy in patients
with IgA nephropathy" written by Li et al. Overall, this is an interesting article that provides additional data regarding the efficacy of tonsillectomy as a treatment for IgA nephropathy with a large cohort of patients. However, I some concerns about the desing and the results of this study that could alter the validity of this results.
Regarding the overall structure of this manuscript there are some paragraphs in the results section that are difficult to understand (lines 182-187 and lines 192-193).
Additionally, in the analysis regarding the interaction with each histological lesion of the MESTC score, I would mention only the most significant results (writting all the lesions such as "When proteinuria was ≤1g/24h and the pathological scores were M0, M1,
E0, S0, S1, T0, T1, C0, C1", makes it very hard to follow).

Experimental design

My main concern regarding this trial is related to the study design. First of all, this is an overall cohort of low-risk IgAN. The authors state that patients with severe forms received steroids or steroids/immunosuppression regimens. However, very few patients have active lesions (overall 6 patients with E1 and 4 patients with C2) and few patients had a proteinuria level over 2 g/day, but almost 60% received some form of IS regimen. I think that the IS may influence the results and this should be acknowledged. I am suggesting that a separate analysis for patients without IS should be undertaken, because in those with IS the validity of the results may be blunted. Additionally, I would recommend that the IS should be included in the factors used for the propensity score.
Additionally, I strongly recommend not ot perform a Kaplan-Meyer analysis for the histological lesions with low frequency (E1, T2, C2).

Validity of the findings

The overall results of this manuscript should be stated clearly. Tonsillectomy had a statistical significance regarding clinical remission or renal outcome only in patients low levels of proteinuria, suggesting that will benefit only in low-risk patients. However, how do these authors interpret their finginds in the light of the recent results regarding the modulation of the gut-association lymphoid tissue in IgAN. I think that a discussion on the contribution of the lymphoid tissue at different MALT sites merits a separate discussion to better integrate these findings.
Additionally, I would revise the part of the manuscript regarding the effect on renal survival. This is a cohort with a low-riskl of progression, followed for a very short time for IgAN and with few renal events. It is hard to state a benefit on renal survival on such a cohort of patients

Reviewer 2 ·

Basic reporting

no comment

Experimental design

no comment

Validity of the findings

This cohort study demonstrated the efficacy and possible indications of tonsillectomy in patients with IgA nephropathy. This manuscript is interesting, but there are some issues that need to be fixed.

1. The definition of negative hematuria and negative proteinuria is not clear. The number of hematuria and proteinuria negatives in the text (L159-162) does not match the number shown in Table 1.

2. There are only 3 patients belonging to E1. Therefore, it is inappropriate to make a statistical analysis of E. I propose to remove E from the statistical analysis.

3. The number of patients with urinary protein 2 or higher is too small for statistical analysis. Thus, I suggest to divide patients into two groups for proteinuria, 1 g/24h or less and >1 g/24h.

4. “mild IgAN” (L34) should be replaced with “early stage IgAN”.

5. The number of patients in each group should be shown in Figure 3-5.

Reviewer 3 ·

Basic reporting

ok

Experimental design

ok

Validity of the findings

some interpretations need corrections

Additional comments

The authors present a retrospective study on the benefits of tonsillectomy in IgA nephropathy. They performed a propensity study, matching individuals with tonsillectomy to controls with a similar risk profile without tonsillectomy. While the study is interesting, there are limitations to address.

The authors state tonsillitis, recurring gross hematuria or chronic tonsillitis were indications of tonsillectomy as an additional treatment, but that it was also offered also to those who wanted it. How many of the 226 had a strict indication of tonsillectomy, and how many just wanted it. This may have significant importance to the outcome, as those who seek more treatments are more likely to be observant and have a more favorable outcome. There is no way to account for this in the propensity method. The authors must detail the indication of tonsillectomy in the results and address this potential bias in the discussion.

In the section “Critical level of proteinuria in the efficacy of tonsillectomy”, the authors should combine the groups 2-3g/day and >3 g1day together as they are underpowered by separating them. Furthermore, they should not conclude that this is only valid in those with low proteinuria, as the lack of association with higher levels is clearly underpowered, not necessarily inefficacious (the graphs favor tonsillectomy in these groups also).

When looking at table 2, it is inappropriate to include « hematuria vs. no hematuria » as a predictor of clinical remission since clinical remission is defined as the absence of both proteinuria and hematuria. The same argument is the same for proteinuria, although proteinuria is expressed as a continuous variable, so it is more acceptable to include it.

The authors should avoid specifying with which pathology score tonsillectomy was beneficial, as the scores where it was not were obviously underpowered. Consider removing lines 221-225 and all other mentions throughout the text, including the abstract. Furthermore, the predictive value of each of these lesions has not been universally accepted, except for the T score. The authors have convincingly shown that the underlining pathology findings do not appear to influence their found benefits with tonsillectomy. They should simplify the reporting of this finding and avoid the clumsy "M0, M1, E0, S0, S1, T0, T1, C0, C1"

Minor comments

Some of the results are presented twice; for example, lines 86-7, and 90-91 should only be in the results section, not in the methods.

figure 2 adds little to figure 3 and should be removed

No need to put decimals when reporting “months.”

Figures 4 and 5 should specify “proteinuria at onset,” not just proteinuria.

To be certain, all of these had a diagnosis of IgA nephropathy before tonsillectomy and all cases were done for the purpose of aiding the outcome of IgA nephropathy.

How was propensity matching performed? Which were the variables included in the determination of the propensity score.

Some of the wording and syntax should be corrected. For example, lines 269 and 270 should not leave data in parenthesis.

---

## Round 0.2 · Major Revisions

Dear authors, please pay careful attention to the concordance between the reported results and conclusions. In addition, it is advisable to be more cautious in terminology, taking into account the study's limitations (which do not allow definitive conclusions).

Reviewer 3 ·

Basic reporting

the syntax should be reviewed throughout the text

Experimental design

nothing to add

Validity of the findings

The authors made some changes to the manuscript that have enhanced it. I have a few remaining questions that warrant changes in the paper's conclusions. Additionally, I have some minor suggestions on wording and presentation.

The authors report information on those who did not receive IS. Could the authors perform the same analyses only on those who did? Without necessarily going through the trouble of drawing all the Kaplan Meier curves, it would be interesting to know this. If the benefits of tonsillectomy are solely driven by those who do not get IS, this would have significant clinical consequences.

The authors have taken the time to combine groups that were too few to conclude. In the different proteinuria groups, the authors state “When proteinuria was >1g/24h, there was no statistical difference between the two groups”. When we look at figures 4 and 5, it appears that the differences between treatment groups in the 1-2 and 2>g/day groups are similar in magnitude to the ≤ 1 g/24h group but underpowered. Perhaps, they should combine the 1-2 to >2 g/day (figures 4 and 5). This is important because the conclusions given in the abstract “it was mainly indicated for patients with early-stage IgAN” are not supported by the data presented and should perhaps be removed.

Furthermore, the authors state “when the pathological damage was mild, the clinical remission rates and the renal survival rates of tonsillectomy group were significantly higher than those of non-tonsillectomy group”. How can the authors say this? There is no evidence of this in the supplemental figures 1 and 2. Tonsillectomy seems as relevant with the presence of S1, T1, and C1.

Given the last 2 remarks, I don’t see how the authors can state in the discussion “The results showed that when proteinuria was ≤1g/24h and the pathological damage was mild, the clinical remission rates and renal survival rates were greater in patients treated with tonsillectomy than without”.

Additional comments

Minor remarks.

The authors state “In China, there is short of studies in which a large cohort of IgAN patients with tonsillectomy are followed to assess the effectiveness of tonsillectomy alone.”, please change the word to “In China, there are few…”

In table 1, no need to put any decimal for µmol of creatinine, blood pressure, and age (e.g. 29.59 years should be simplified to 30). No need to put two decimals for % above 10% (68.14% should be left as 68%). When giving 95% CI, please change (0.84to1.22) to (0.84-1.22)

Could the authors define “chronic tonsillitis”?

The authors state “and 6 just wanted it to obtain clinical remission”, perhaps they should change it “6 had no documented tonsil anomaly”

The authors state “while tonsillectomy was independently beneficial for renal survival”. The authors should say was “independently associated with a greater renal survival”. The previous way it is written infers causality.

The authors state “was no difference for patients with and without tonsillectomy”. Please change the text to “was not different”

The authors state “Considering the validity of statistics, we did not perform Kaplan-Meyer analysis on histological lesions with low frequency (E1, T2, C2).” They should remove the “considering the validity of statistics”

The authors should avoid using “Nephrologically” in the abstract and discussion.

---

## Round 0.3 · accepted · Accept

As the authors made changes in accordance with suggestions, including new statistical analysis, I consider their answers satisfactory and I view no need for a new round of review.

The authors acceptable resolved the previous concerns. No further comments.